# Stochastic modeling of intra- and inter-hospital transmission in Middle East respiratory syndrome outbreak

**Youngsuk Ko[1], Jacob Lee[2], Eunok Jung** [3]*

**1** Institute of Mathematical Sciences, Konkuk University, Seoul, Republic of Korea, **2** Division of Infectious Diseases, Department of Internal Medicine, Kangnam Sacred Heart Hospital, Hallym University College of Medicine, Seoul, Republic of Korea, **3** Department of Mathematics, Konkuk University, Seoul, Republic of Korea

\* junge@konkuk.ac.kr

**Data availability statement:** The dataset and code used in this study are publicly available in

## Abstract

Middle East Respiratory Syndrome (MERS) is an endemic disease that presents a significant global health challenge characterized by a high risk of transmission within healthcare settings. Understanding both intra- and inter-hospital spread of MERS is crucial for effective disease control and prevention. This study utilized stochastic modeling simulations to capture inherent randomness and unpredictability in disease transmission. This approach provides a comprehensive understanding of potential future MERS outbreaks under various scenarios in Korea. Our simulation results revealed a broad distribution of case number, with a mean of 70 and a prediction interval of [0, 315]. Additionally, we assessed the risks associated with delayed outbreak detection and investigated the preventive impact of mask mandates within hospitals. Our findings emphasize the critical role of early detection and the implementation of preventive measures in curbing the spread of infectious diseases. Specifically, even under the worst-case scenario of late detection, if mask mandates achieve a reduction effect exceeding 55%, the peak number of isolated cases would remain below 50. The findings derived from this study offer valuable guidance for policy decisions and healthcare practices, ultimately contributing to the mitigation of future outbreaks. Our research underscores the critical role of mathematical modeling in comprehending and predicting disease dynamics, thereby enhancing ongoing efforts to prepare for and respond to MERS or other comparable infectious diseases.

## Introduction

Middle East respiratory syndrome (MERS) results from infection with the Middle East respiratory syndrome coronavirus (MERS-CoV) [1], a zoonotic pathogen capable of transmission between animals and humans. Clinically, MERS patients exhibit varying degrees of symptoms, including fever, cough, and shortness of breath [2]. Disease progression leads to complications such as pneumonia, acute respiratory distress syndrome, kidney failure, and multiorgan dysfunction. The average fatality rate stands at approximately 36%, with regional

the figshare repository at DOI:
10.6084/m9.figshare.26317555.v3 (https://
doi.org/10.6084/m9.figshare.26317555.v3).
There are no restrictions on data access.

**Funding:** This research was supported by the
Government-wide R&D Fund Project for
Infectious Disease Research (GFID), Republic of
Korea (grant No. HG23C1629). This paper is
supported by the Korea National Research
Foundation (NRF) grant funded by the Korean
government (MEST)
(NRF-2021R1A2C100448711). The funders had
no role in study design, data collection and
analysis, decision to publish, or preparation of
the manuscript.

**Competing interests:** The authors have
declared that no competing interests exist.

variations ranging from 14% to 44% [3–5] based on reported cases. MERS was initially identified in Saudi Arabia in 2012 [2] and continues to sporadically emerge in endemic regions, notably Saudi Arabia, which reports the highest case count [5]. As of 2024, Saudi Arabia has documented 2,204 cases, resulting in 862 deaths [5].

Subsequent to Saudi Arabia, South Korea reported the second highest incidence of MERS infections, predominantly linked to a single outbreak. In 2015, Korea faced a MERS outbreak originating from an individual who had recently visited the Middle East [6,7]. The disease primarily disseminated within hospital settings, resulting in 186 confirmed cases and 36 fatalities—a significant public health challenge for the country [8]. Notably, this outbreak exhibited a pronounced risk of transmission within hospitals, while general community spread remained limited [9,10]. These dynamics emphasize the critical need to comprehend both intra- and inter-hospital MERS transmission pathways for effective disease management and prevention.

Mathematical modeling plays a crucial role in investigating infectious diseases, offering a structure approach for deciphering and forecasting transmission dynamics [11]. Key advantages of such models lie in their ability to yield quantitative insights. By capturing intricate interactions among hosts, pathogens, and the environment, these models facilitate outbreak prediction, intervention assessment, and informed public health policy. Unlike decisions based on general observations, these models rely on precise numerical data. Policymakers benefit from these accurate data when evaluating potential interventions, such as vaccination campaigns, travel restrictions, and enforcing social distancing measures [12–14]. Specifically for MERS, mathematical models have been pivotal in uncovering transmission patterns, predicting outbreaks, and identifying effective public health measures [15,16]. Assessments have highlighted the potential for sustained MERS transmission, emphasizing hospital-based spread [17,18]. Investigating superspreading events relative to the basic reproductive number has been pivotal [19,20]. Agent-based modeling techniques illuminate the impact of superspreading events on the MERS transmission dynamics of MERS, particularly during the Korean outbreak [21].

In this study, we utilize stochastic modeling simulations to capture the inherent randomness and unpredictability of disease spread, encompassing both intra- and inter-hospital transmission dynamics. Our model yields a comprehensive understanding of future MERS outbreaks across diverse scenarios, informing effective control strategies in South Korea. Our investigation focused on Gangnam District in Seoul, Korea, which comprises two tertiary referral hospitals (average bed count: 1,533), two general hospitals (average bed count: 225), and 32 smaller hospitals (average bed count: 73) [23]. The local community population stands at 650,000 [24]. Notably, our analysis excluded smaller clinics. Additionally, we assess the risk associated with delayed outbreak detection and investigate the preventive impact of mask mandates within healthcare facilities. These considerations are important given the current global health landscape, where timely detection and preventive measures are essential in managing infectious disease spread. Our research aims to contribute to preparedness and response efforts for MERS and related infectious diseases, offering valuable insights to guide policy decisions and healthcare practices, ultimately mitigating future outbreaks.

## Materials and methods

### Stochastic modeling of MERS outbreak

Our model includes individuals who are either hospitalized or directly interacting with hospital settings in the Gangnam District, Seoul. Specifically, the included populations are inpatients, hospital visitors, and medical staff across 36 hospitals (two tertiary, two general, and 32

smaller hospitals). The simulation is initiated with the importation of a primary case into the local community (population: 650,000), and transmission dynamics are limited to hospital-related settings. We excluded small clinics that do not accommodate inpatients, due to lack of transmission data and limited epidemiological relevance. Additionally, we did not include individuals from the general community who had no hospital exposure during the outbreak.

We analyzed the following key outcome variables and the role of uncertainty-from the stochastic simulations:

- Total number of individuals who became infected during the simulated outbreak.
- Number of hospitals with at least one infected individual (i.e., where any exposed, infectious, hospitalized, or isolated individual is present).
- Time elapsed from symptom onset of the index case to the isolation of the last infected individual.
- Maximum number of individuals in the isolated compartment at any single point in time.
- Daily number of individuals in exposed, infectious, and isolated states throughout the simulation.
- Sensitivity analysis metric to quantify the influence of input parameters on the total number of infections.

Drawing upon the Susceptible-Exposed-Infectious-Recovered (SEIR) mathematical model, we constructed a framework that accounted for both intra- and inter-hospital transmission of MERS. Within our model, hosts traverse six epidemiological stages: susceptible ($S$), exposed ($E$), infectious ($I$), hospitalized ($H$), isolated ($Q$), and recovered ($R$). In the hospital context, hosts are further categorized into medical staff (subscript $M$), inpatients ($P$), and visitors ($V$). Asymptomatic cases were not separately modeled but were implicitly represented within either the infectious or hospitalized states. Superspreaders were not explicitly identified in order to reflect generalized transmission patterns rather than outlier-driven dynamics. Importantly, we did not differentiate superspreaders in this study. Fig 1 depicts the flowchart of our epidemiological model, illustrating disease progression stages. Solid arrows denote disease transmission pathways. The terms $\Lambda_M^i$, $\Lambda_P^i$, and $\Lambda_V^i$ represent the force of infection for nosocomial infections, influenced by $I_M^i$, $I_P^i$, $I_V^i$, and $H^i$. Here, the superscript $i$ denotes hospital identifiers. Additionally, $\Lambda$ represents the force of infection for local spread, with contributions from $I$, $I_M^i$, and $I_V^i$. We assume a Markovian process for disease transmission, where future states depend solely on the current state, independent of preceding events. Dashed arrows with numerical labels indicate delayed reactions (non-Markovian process) in disease progression. To simulate disease spread, we aggregate recorded time delays and incorporate them into the model [25]. Transmission rates and reductions in intra-hospital transmission due to non-pharmaceutical interventions (NPIs) are estimated based on the Pyeongtaek St. Mary's Hospital outbreak, the site of the initial hospital cluster infection [26]. We employed stochastic simulation based on a modified Gillespie algorithm, incorporating both Markovian (non-delayed) and non-Markovian (delayed) processes. Each simulation was initiated with the introduction of a single index case into the local community. For each scenario, we performed 10,000 independent simulation runs to capture the stochastic variability in transmission dynamics. Detailed description is in S1 Text [22].

## Model simulation scenarios

We designed a series of simulation scenarios to assess outbreak dynamics under varying conditions. In the baseline scenario we assume 0% mask effectiveness—reflecting the low

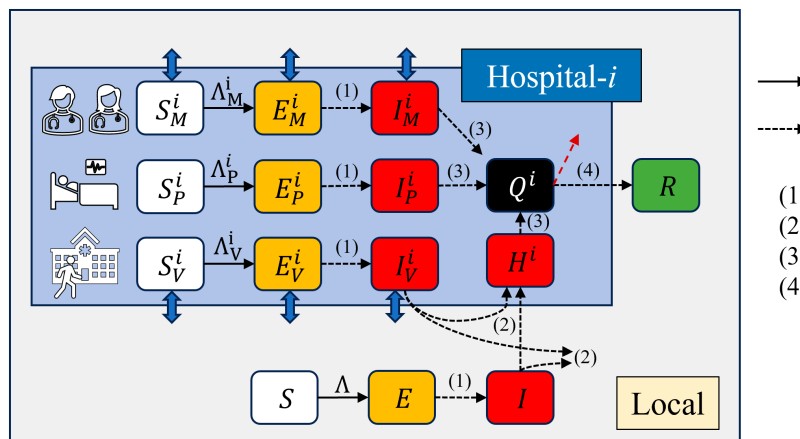

**Fig 1. Flow diagram of Middle East Respiratory Syndrome transmission model considering intra- and inter-hospital transmission.**

awareness and minimal mask use in Korean hospitals during the 2015 outbreak—while the observed reduction in intra-hospital transmission is modeled solely through post-recognition interventions. The scenarios are described below in sequential order:

1. **Baseline scenario**: The index case is introduced into the community, and the outbreak is recognized 5 days after hospital admission. Hospital visits are prohibited at recognition, and intra-hospital transmission is reduced by 41.02% (S1 Text) [8]. Upon symptom onset, a host is randomly admitted to a hospital.
2. **Outbreak detection delay**: We varied the detection delay from 1 to 28 days to evaluate the risk of late outbreak recognition.
3. **Mask intervention**: We simulated the impact of intra-hospital mask mandates, assuming reduction effects in transmission ranging from 0% (baseline) to 80%.

Fig 2 shows the simulation flow, encompassing outbreak recognition and intervention application. Once the outbreak is identified, we implement NPIs: hospital visits are prohibited (i.e., $S_V^i$ transitions to susceptible ($S$)), and the intra-hospital transmission rates decrease by 41.02%. The estimation of this reduction is detailed in S1 Text. Additionally, beyond the baseline scenario, we explore additional scenarios considering the following factors:

- Delay in outbreak detection: The risk associated with outbreak recognition delay is investigated, considering a range from 1 to 28 days. Notably, for the baseline scenario, this delay is specifically quantified as five days.
- Mask mandates in hospital: In hospital settings, the preventive impact of mask-wearing intervention is assessed accounting for mask type and enforcement. We assumed that implementing mask mandates would lead to a decrease in intra-hospital transmission rates. The resulting transmission reduction effect ranges from 0% (baseline scenario) to a maximum of 80% [27].

Not that based on Offeddu et al.'s meta-analysis (risk ratios 0.13–0.59, corresponding to 41–87% reduction in infection), we set the maximum mask effectiveness at 80%.

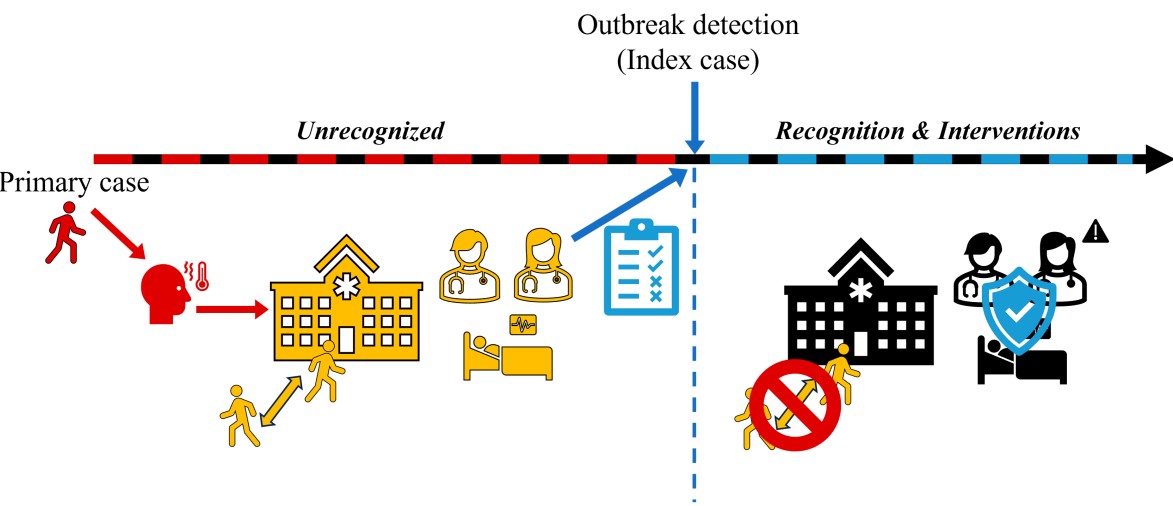

**Fig 2. Flow of the simulation from the import of the primary case to the outbreak recognition and reactive interventions.**

Lastly, we performed a sensitivity analysis to identify the key factors influencing the outbreak. Using Latin Hypercube Sampling, we quantified the Partial Rank Correlation Coefficient (PRCC) values [29]. Our model incorporated several input parameters: outbreak detection timing ($\tau_{rec}$), intra-hospital transmission rates ($\beta_H$), intra-hospital infectious period for the primary case ($\tilde{\tau}_{I \to Q}$), intra-hospital infectious period excluding the primary case ($\tau_{I \to Q}$), local transmission rate ($\beta_L$), local infectious period for the primary case ($\tilde{\tau}_{I \to H}$), local infectious period excluding the primary case ($\tau_{I \to H}$), incubation period ($\tau_{E \to I}$), and the number of hospital visitors ($S_V^i$). We performed 200,000 simulation runs, introducing a uniform variation of $\pm 30\%$ based on default baseline values. Notably, $\beta_H$ encompassed all transmission rates within the hospital. Additionally, we accounted for delay-inputs by adjusting the variation ratio using values generated during the model simulation. The output of the model represented the cumulative number of infections, capturing the transition from susceptible to exposed stage.

## Results

We examined multiple scenarios to evaluate the impact of outbreak detection timing and hospital-based preventive interventions on nosocomial transmission dynamics. Specifically, we compared a baseline scenario—representing moderate early detection and no mask mandates—with alternative scenarios involving delayed detection (1 to 28 days), mask mandates with varying effectiveness (0% to 80%), and their combined effects. Each scenario was simulated using a stochastic framework to assess variability in outbreak size, duration, and peak burden.

### Baseline scenario simulation and sensitivity analysis

In Fig 3, panels A, B, and C depict the distributions of confirmed cases, exposed hospitals, and outbreak duration, respectively. We define an exposed hospital as one that has at least one infected host ($E$, $I$, $H$). Outbreak duration is defined as the time from symptom onset of the index case to the isolation of the last infected host. Among the simulation runs, approximately 8% conclude without secondary infections. The mean number (95% prediction interval; PI)

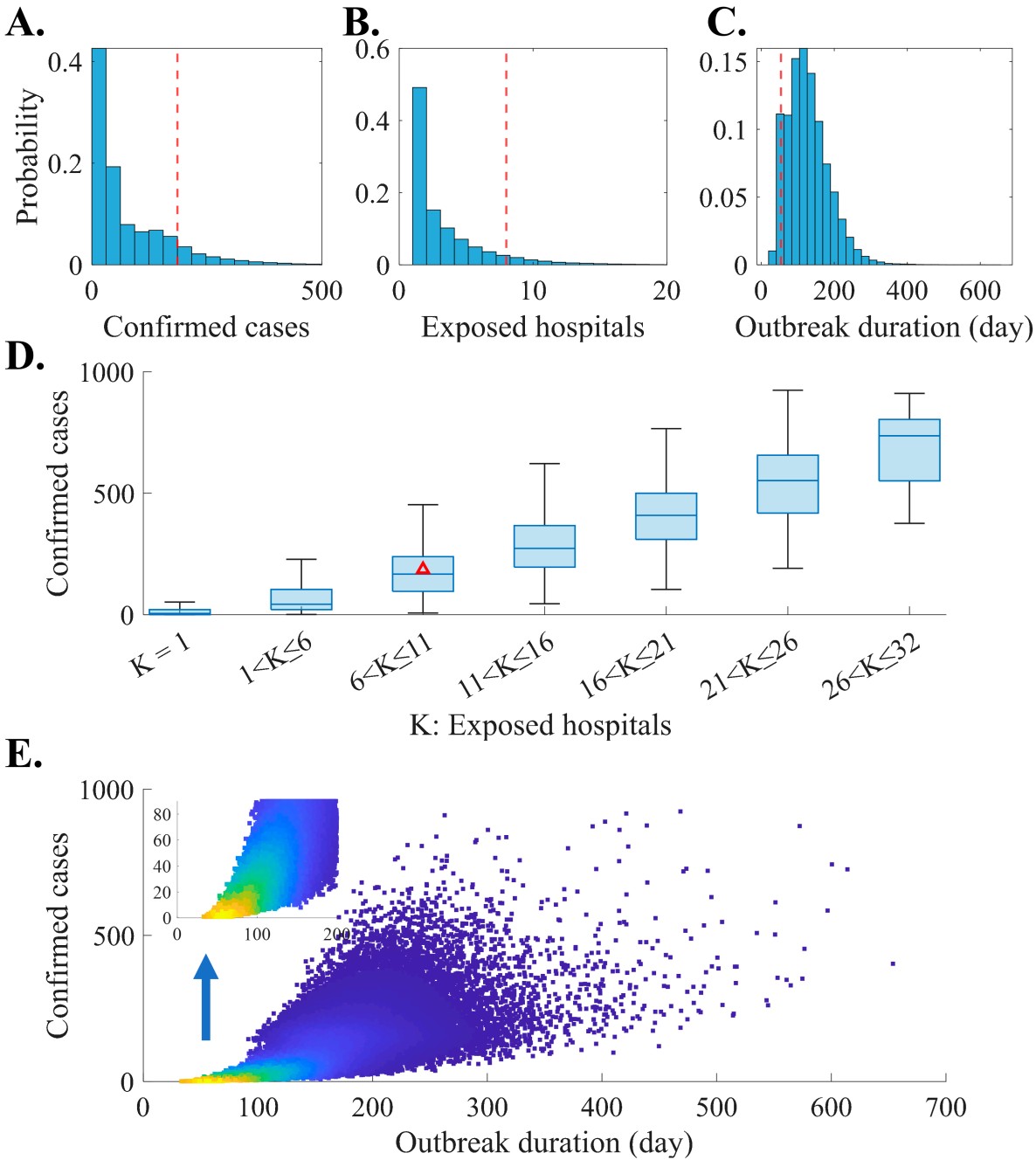

**Fig 3. Baseline simulation results of outbreak outcomes: Distribution of the number of confirmed cases, exposed hospitals, and outbreak duration (A, B, and C), Distribution of confirmed cases using a box-chart graph according to the range of exposed hospitals (D), Weighted scatter plot of the number of confirmed cases according to the outbreak duration (E).** From panels A to C, red dashed vertical lines indicate the actual outbreak outcome during the 2015 MERS outbreak in Korea, which is also marked using red triangle in a panel D.

for confirmed cases, exposed hospitals, and outbreak duration are 70 ([0,315]), 3 ([1,12]), and 119 ([33,253]), respectively. Panel D presents a box-chart graph combining panels A and B, with a red asterisk marking the MERS outbreak in Korea in 2015 (case number: 186, exposed

hospitals: 8). In the cases limited to the Gangnam region, there were 83 newly infected individuals at a single hospital (Samsung Medical Center), not including those who were transferred after being infected elsewhere [6,7]. To quantify the relationship between the total number of confirmed cases and the number of exposed hospitals across simulation runs, we calculated Pearson's correlation coefficient [28]. Notably, a strong correlation exists between confirmed cases and exposed hospitals (coefficient: 0.70, p-value approximately zero). Similarly, panel E presents a scatter plot depicting the relationship between the number of confirmed cases and outbreak duration. The color scale indicates clustering, with samples closer to yellow representing stronger clustering. A correlation analysis yields a coefficient of 0.77, smaller than the previously estimated coefficient. Additionally, the associated p-value is close to 0.

Fig 4 shows the time series graphs of infected hosts. Panels A, B, and C correspond to the number of hosts in the exposed, infectious, and isolated states, respectively. The black curve represents the mean value, whereas the red area indicates the 95% PI. Approximately 60 days after the initial onset, the exposed and infectious stages peak mean values of 6 and 4, with 95% PI of [0,29] and [0,22], respectively. Notably, the number of isolated patients could exceed 100 considering the 95% PI.

Fig 5 illustrates the time-dependent PRCC values for each input. $\beta_H$ consistently exhibits the highest PRCC, ranging from 0.16 to 0.31 throughout the observation period. Following closely are $\tilde{\tau}_{I \to Q}$ (PRCC: [0.07, 0.12]) and $\tau_{I \to Q}$ (PRCC: [0.01, 0.21]). During the initial 15 days after onset, $\tilde{\tau}_{I \to Q}$ exhibits higher PRCC; however, $\tau_{I \to Q}$ surpasses it subsequently. The parameter $\beta_L$ has the second-highest PRCC values within the range [0.04, 0.08]. Notably, 87 days after onset, the PRCC of $\beta_L$ exceeds that of $\tilde{\tau}_{I \to Q}$. Interestingly, $\tau_{E \to I}$ is the sole input among all factors where the sign of PRCC changes from positive to negative. Initially positive (maximum 0.03), PRCC transitions to negative (minimum -0.02) 13 days after the outbreak began. Additionally, $\tau_{I \to H}$ and $S_V^i$ exhibited PRCC of 0.01 to 0.02 and –0.01 to 0.02, respectively, indicating a negligible effect on outbreak outcomes.

## Risk of late detection and preventive impact of mask-wearing in hospital

To assess the effects of delayed outbreak recognition and hospital-based preventive interventions, we simulated additional scenarios varying the detection delay and the effectiveness

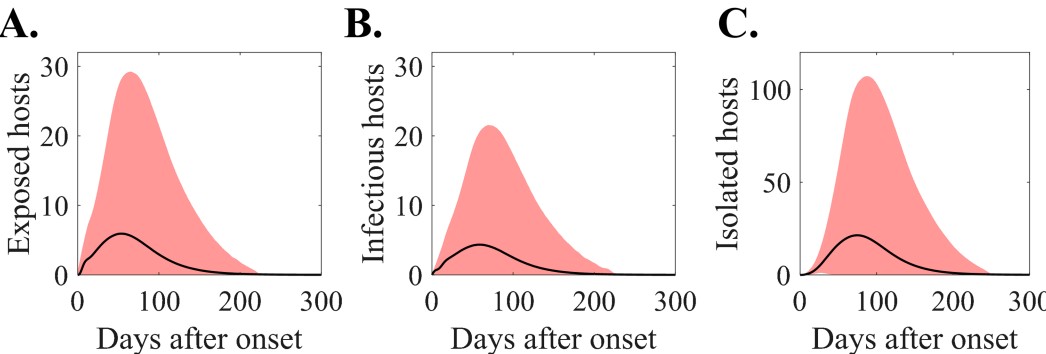

**Fig 4. Number of infected hosts in different stages: Disease-exposed (A), Infectious (B), Isolated (C).**

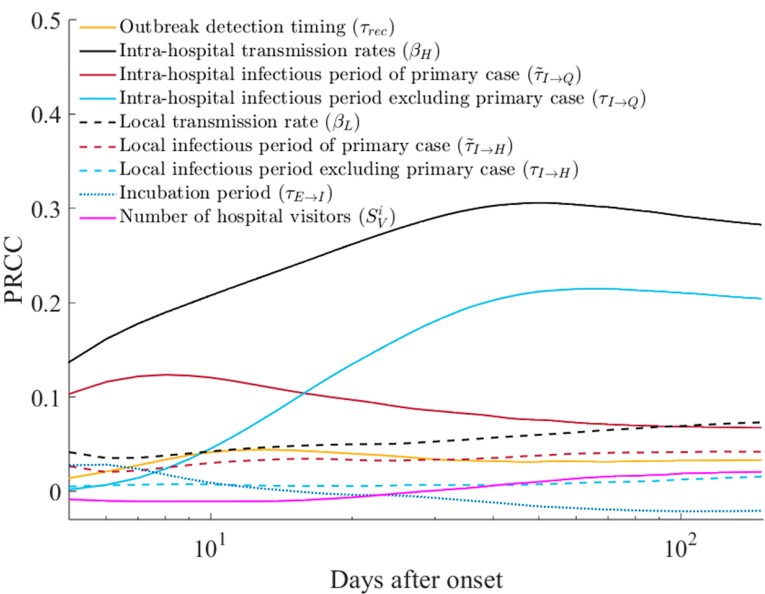

**Fig 5. Results from the parameter sensitivity analysis, quantified using the Partial Rank Correlation Coefficient (PRCC).** Notably, the X-axis (time, days after onset) is log-scaled.

of mask mandates. Specifically, we analyzed the peak number of isolated patients —a critical risk indicator—by simultaneously adjusting the outbreak detection delay and the effectiveness of mask mandates within hospitals. Fig 6 illustrates the results using filled contour plots, depicting the mean value (panel A) and the maximum value within the 95% PI (panel B). When mask mandates have no effect (represented by a value of 0 on the X-axis), the peak number of isolated patients can exceed 350 within the 95% PI if outbreak detection is delayed by up to 28 days. However, with the fastest detection (one day), the peak is expected to average approximately 25 and reach up to 125 within the 95% PI. When mask mandates have an effect exceeding 40%, the peak number of isolated patients—considering outbreak detection at its latest (28 days)—is smaller than the peak without mask mandates when comparing maximum values within the 95% PI. Furthermore, if the effect of mask mandates exceeds 70%, the maximum number of peak isolated patients within the 95% PI under late outbreak detection, is smaller than the mean when detection is fastest without mask mandates. The contour line representing the mean peak number of isolated patients at 50 is of particular significance. Without the effect of mask mandates, this threshold is achieved with a 12-day detection delay. When extrapolated to the far right, the contour line corresponds to a scenario involving a 28-day detection delay and a 30% impact from mask mandates.

**Summary of scenario comparisons.** Compared to the baseline scenario, delayed outbreak detection led to substantial increases in outbreak size and isolation burden. However, when moderate to high mask effectiveness (≥60%) was assumed, these effects were largely mitigated. The combined implementation of early detection and mask mandates was the most effective in reducing peak isolation burden. Overall, these results emphasize the critical role of timely outbreak recognition and intra-hospital interventions in minimizing outbreak impact.

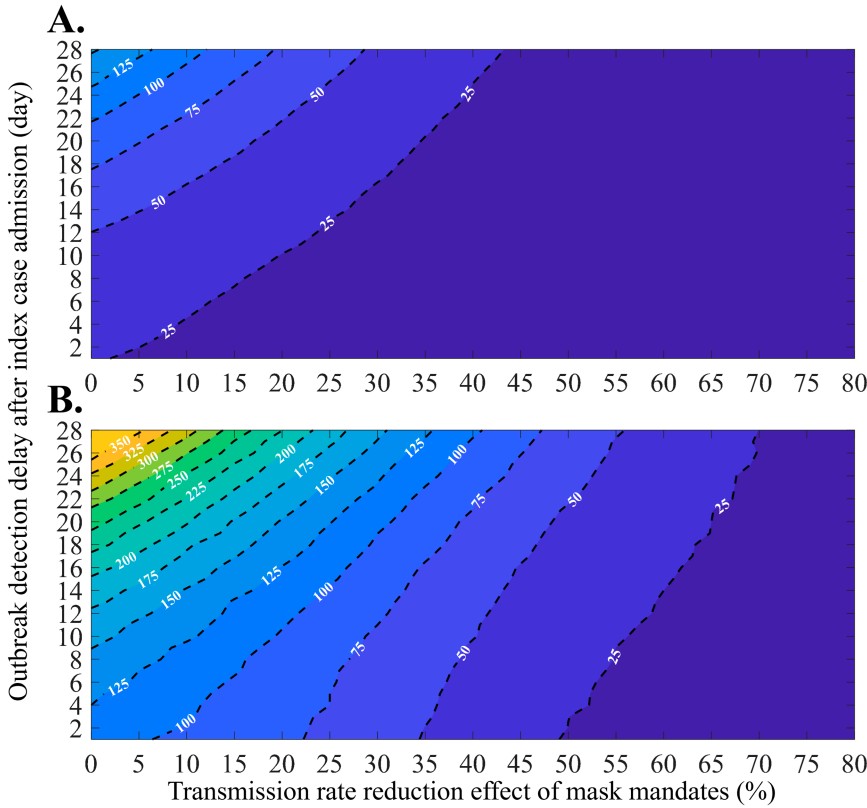

**Fig 6. Filled contour plots depict the peak number of isolated patients as a function of changes in the transmission reduction effect due to mask mandates and outbreak detection delay: Mean value (A), and Maximum value within the 95% PI (B).**

## Discussion

The baseline scenario simulation results revealed a broad distribution of case numbers, ranging from 0 to 315 within the 95% PI. Additionally, the simulation indicated a high probability of small-scale outbreaks (Fig 3A). This phenomenon arises because an outbreak initiated by a single case is significantly influenced by the individual events, potentially leading to early containment or limited transmission to the second or third generation of infection. Notably, the number of confirmed cases observed during the 2015 Korean fell slightly below the upper bound of the PI for the baseline scenario (upper 10%). When considering only the cases limited to the Gangnam District, the number falls within the upper 29% of the PI. Furthermore, when we further analyzed the simulation results by considering the number of hospitals exposed to the disease, the distribution of confirmed cases could vary (Fig 3D). In the 2015 simulation results, which align with the observed number of exposed hospitals (nine hospitals), the case represents the upper 44% of the distribution. Therefore, we can conclude that the 2015 scenario is not an unrealistic situation. Additionally, Fig 4 illustrates the estimated number of infected hosts at a specific time. This information serve as a reference for establishing tracking targets during contact tracing, particularly for epidemiological investigators, at the outset of an outbreak. For instance, 20 days after the primary case onset, the mean number of exposed and infectious hosts is 3 and 2 (with 95% PI [0,10] and [0,7]), respectively. However, there was a difference in the trend between our simulation results and the actual

2015 case, particularly in the outbreak duration. The simulation results showed an average outbreak duration of 119 days, whereas the actual outbreak in 2015 lasted for 54 days [8]. This difference arises from the characteristics of the model used in this study. The actual phenomenon was driven by a small number of superspreaders, but this study averaged the effects of all spreaders in the simulation.

The sensitivity analysis conducted on the baseline scenario yielded quantitative insights (Fig 5). Notably, the infectious period of the initial primary case was found to be more significant than that of the remaining infected individuals during the early phase of the outbreak, as revealed by the time-varying PRCC. In our investigation, we noted that the local community's transmission rate significantly influenced the outbreak scale. This observation raises the question of whether NPIs are also warranted at the local community level. However, considering the socio-economic burden associated with implementing NPIs locally —a different scale from the hospital-level control measures—the results of our sensitivity analysis paradoxically underscores the importance of early detection of the index case (primary case) rather than immediate NPI application within the local community. Notably, during the COVID-19 pandemic, several countries, including Korea, faced a substantial increase in NPIs burden when the outbreak reached a nationwide scale [30,31].

Our simulation results reveal that rapid detection within hospitals can significantly mitigate outbreak scale, while delayed detection exacerbates the situation (Fig 6B). The peak size of isolated patients emerges as a critical factor affecting medical resources; exceeding local community capacity can lead to rapid deterioration [32]. In Gangnam district (2024), where 77 isolation beds are available [23], a detection delay of more than 18 days —without preventive mask mandates —could result in medical collapse. Our findings underscore the need for a constant level of isolation ward capacity and provide a realistic risk assessment (.g., a four-day detection delay after index admission corresponds to an upper-bound estimate of approximately 125 cases within the 95% PI).

One strength of this study is its flexible framework for evaluating multiple policy scenarios using stochastic modeling that reflects real-world delays and heterogeneity. Our use of epidemiologically-derived parameters and detailed simulation design enables realistic assessment of hospital-level transmission dynamics and control strategies. The limitations of this study are as follows: (1) The transmission rate within the hospital can be heterogeneous depending on the hospital environment; however, in this study, the parameters estimated from the Pyeongtaek St. Mary's Hospital case were applied uniformly across all hospitals. (2) Unreported cases or superspreaders could not be distinguished. This resulted in a longer outbreak duration compared to the real outbreak case. (3) The phenomenon of doctor shopping, where the same patient visits multiple hospitals, was not considered [36]. (4) Patients transfer between hospitals and maximum isolation spaces capacity were also omitted. Future studies should address these limitations. (5) We excluded small clinics because they lack inpatient data and modeling thousands of tiny facilities exceeds our stochastic framework; we will address this in future agent-based work.

## Conclusion

Our model focuses on both of inter- and intra-hospital transmission dynamics. Reflecting the minimal mask use and active visitation culture during the 2015 Korean MERS outbreak, we quantify how hospital-based mask madates can both curb nosocomial spreads and avert broader community outbreaks. Our findings highlight the critical role of early outbreak recognition and targeted hospital-level interventions in controlling nosocomial MERS transmission. Even under delayed detection, effective mask mandates can significantly

reduce peak burden, underscoring their importance as a first-line defense in healthcare settings.

## Recommendations

Based on our findings, we recommend prioritizing early outbreak recognition through streamlined diagnostic workflows in hospitals. Our simulation results indicate that even modest delays in detection can substantially increase outbreak size and isolation burden. In particular, detection delays beyond two weeks may exceed local isolation capacity unless mitigated by additional interventions.

We also recommend maintaining mask mandates within hospital settings during periods of emerging respiratory threats. Our results show that mask effectiveness of 60% or higher can partially compensate for delayed detection and significantly reduce peak isolation burden. Finally, maintaining a baseline level of surge capacity for isolation beds in densely populated urban areas may help buffer against sudden outbreak escalation.

## Supporting information

**S1 Text. Supplementary appendix.** This contains detailed explanation of methods for the research.
(PDF)

## Author contributions

**Conceptualization:** Youngsuk Ko, Eunok Jung.

**Data curation:** Youngsuk Ko.

**Formal analysis:** Youngsuk Ko, Jacob Lee, Eunok Jung.

**Funding acquisition:** Eunok Jung.

**Investigation:** Youngsuk Ko, Jacob Lee, Eunok Jung.

**Methodology:** Youngsuk Ko.

**Project administration:** Eunok Jung.

**Resources:** Youngsuk Ko.

**Software:** Youngsuk Ko.

**Supervision:** Eunok Jung.

**Validation:** Youngsuk Ko, Jacob Lee.

**Visualization:** Youngsuk Ko.

**Writing – original draft:** Youngsuk Ko.

**Writing – review & editing:** Youngsuk Ko, Jacob Lee, Eunok Jung.

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
