## [Decision Letter · Decision Letter 0]

5 May 2025

PONE-D-25-03162Stochastic Modeling of Intra- and Inter-Hospital Transmission in Middle East Respiratory Syndrome OutbreakPLOS ONE

Dear Dr. Jung,

Thank you for submitting your manuscript to PLOS ONE. After careful consideration, we feel that it has merit but does not fully meet PLOS ONE’s publication criteria as it currently stands. Therefore, we invite you to submit a revised version of the manuscript that addresses the points raised during the review process.

We look forward to receiving your revised manuscript.

Kind regards,

Eunha Shim

Academic Editor

PLOS ONE

Journal Requirements:

3. Please note that your Data Availability Statement is currently missing the repository name. If your manuscript is accepted for publication, you will be asked to provide these details on a very short timeline. We therefore suggest that you provide this information now, though we will not hold up the peer review process if you are unable.

Reviewers' comments:

Reviewer's Responses to Questions

**Comments to the Author**

1. Is the manuscript technically sound, and do the data support the conclusions?

Reviewer #1: Yes

Reviewer #2: Partly

Reviewer #3: Yes

Reviewer #4: Yes

2. Has the statistical analysis been performed appropriately and rigorously? 

Reviewer #1: Yes

Reviewer #2: No

Reviewer #3: Yes

Reviewer #4: Yes

3. Have the authors made all data underlying the findings in their manuscript fully available?

Reviewer #1: Yes

Reviewer #2: Yes

Reviewer #3: Yes

Reviewer #4: Yes

4. Is the manuscript presented in an intelligible fashion and written in standard English?

Reviewer #1: Yes

Reviewer #2: Yes

Reviewer #3: Yes

Reviewer #4: Yes

5. Review Comments to the Author

Reviewer #1: - Introduction:

The background related to Gangnam District in Seoul, Korea, and the two surveyed hospitals should be mentioned here (population study)

- Methods:

o Add information on the sampling methodology

o Inclusion/exclusion criteria

o List and define variables to be analyzed

o The various scenarios and their sequence should be indicated, from baseline to other models

o The sensitivity analysis to be mentioned as well

- Results:

The results for each model should be described more systematically, and summarized. What has changed between the baseline model and the other model?

- Discussion:

Each result should be discussed more systematically and following the order of appearance in the results section

- Strengths and Limitations of this study should be indicated separately

- A section on Conclusion is to be added separately

- A separate section on Recommendations to be added as well

Reviewer #2: Reviewer report

Manuscript: PONE-D-25-03162

Title: Stochastic Modeling of Intra- and Inter-Hospital Transmission in Middle East

Respiratory Syndrome Outbreak

Summary:

The study used stochastic modeling simulation approaches to understand MERS transmission dynamics with inherent randomness and unpredictability, with hope that the outcomes will guide disease management policy decisions. The methods as described are too shallow and need to be more detail as this is very important in reporting modeling work to ensure reproducibility of the study. Some few sentences need to be rephrased as well. Some of the content is misplaced e.g. some stuff suitable for materials and methods is wrongly introduced in the results section e.g. under sensitivity analysis.

Major observation:

1. The materials and methods are too shallow and hard to comprehend. Introduce key steps in this section and describe them concisely. For example, the statistical methods used in assessing correlations between correlation exists between confirmed cases and exposed hospitals need to be described among others.

2. In materials and methods, you mention that your analysis ignored smaller clinics. Much as this could be a simplifying assumption, it may impact the study outcomes. Investigate (as a sensitivity analysis) how their inclusion would affect the study outcomes.

3. For improved content flow, some key informative details on model building and parameterization need to be (concisely) introduced in the main text instead of supplementary material. E.g., in line 83, on how was the value of 59% arrived? This and other similar mentions need description instantly. You can keep the detailed descriptions about the same parameters in the supplementary file.

4. How realistic is the assumption of 0% mask use impact in the baseline scenario? It is a known fact that irrespective of the prevailing circumstances, some people would be using masks anyway. This need to be revised.

5. What informed your choices for the ranges explored in the alternative scenarios explored? Justification should be added to the main text and practicality and feasibility should be key. You cite [28] on 80% the second scenario, (concisely and briefly) describe how they came up with that number in the main text.

6. The code used for the sensitivity analysis should be provided as part of the supplementary information files.

7. Rearrange your contact to ensure that content appears where it is most suited. Case in point is the description on sensitivity analysis (in lines 121-132) that you introduce in the results section yet it is methods.

Some of the minor observations:

8. Tense and grammar edits are needed e.g. in abstract where you write “Our findings emphasized…” shouldn’t it be “emphasize”?

9. Line 14: replace “singular” with “single”

10. Line 47: delete “of”

11. Under materials and methods, separate data description and model description in the first subheading.

12. Line 68: wrong phrasing in “To simulate our model…”. It is not the model that you are simulating rather it is the disease spread that you are simulating using the model

13. Line 97: use “distribution” instead of “distributions”?

14. Check and revise the flow of the sentence in line 98.

15. Line 146: what are those two factors? Mention them explicitly or improve sentence flow.

Reviewer #3: 1. The authors must provide the complete code to reproduce all model fitting procedures and simulation output presented in the manuscript.

2. There is already a substantial modeling study of the MERS. What is the unique contribution of this work, and how does the model offer meaningful insights or practical relevance in the current context?

3. The study assumed that the number of hospital visitors equals to the number of inpatients. Is there a reference to support this assumption? In addition, a sensitivity analysis exploring how variations in the ratio of hospital visitors to inpatients affect the model outcomes should be conducted and included.

4. In Figure 1, which presents a diagram of the model, the solid arrows represent the pathways of the infection transmission. However, it is unclear what the dotted arrows and red dotted arrows indicate. Additionally, both I_V^i and I flow into “red (2)”, but it is unclear what this “red (2)” signifies. Is it not a specific compartment? Why was it depicted in red?

5. The equations in the Supporting Information (S1 text) need to be organized, as some equations are repeated. Furthermore, based on the diagram in Figure 1, I_M^i, I_P^i, and H^i flow into Q^i. However, the equations state (dQ^i)/dt=I_M^i (t-τ_(I→Q) )+I_P^i (t-τ_(I→Q) )+I_V^i (t-τ_(I→Q) )-Q^i (t-τ_(Q→R)). The author could provide a more detailed explanation or clarification.

Reviewer #4: The article “Stochastic Modeling of Intra- and Inter-Hospital Transmission in Middle East Respiratory Syndrome Outbreak” models the spread of MERS within hospitals and the general population in the Gangnam District of Seoul, Korea. The authors use the model to predict outputs such as the size of the outbreak, number of exposed hospitals, and outbreak duration and assess the effect of mask mandates on the results. I thought the article was very well written, the methodology sound, and interpretation reasonable. My comments are generally minor. Please see the attached document for the comments.

6. PLOS authors have the option to publish the peer review history of their article (what does this mean?). If published, this will include your full peer review and any attached files.

Reviewer #1: No

Reviewer #2: **Yes: **Amos Ssematimba

Reviewer #3: No

Reviewer #4: No

---

## [Author Response · Author response to Decision Letter 1]

14 Jul 2025

Dear Reviewers,

Thank you for insightful comments. We have revised paper and attached the response as word file.

---

## [Decision Letter · Decision Letter 1]

27 Aug 2025

Stochastic Modeling of Intra- and Inter-Hospital Transmission in Middle East Respiratory Syndrome Outbreak

PONE-D-25-03162R1

Dear Dr. Jung,

We’re pleased to inform you that your manuscript has been judged scientifically suitable for publication and will be formally accepted for publication once it meets all outstanding technical requirements.

Kind regards,

Eunha Shim

Academic Editor

PLOS ONE

Additional Editor Comments (optional):

Reviewers' comments:

Reviewer's Responses to Questions

**Comments to the Author**

1. If the authors have adequately addressed your comments raised in a previous round of review and you feel that this manuscript is now acceptable for publication, you may indicate that here to bypass the “Comments to the Author” section, enter your conflict of interest statement in the “Confidential to Editor” section, and submit your "Accept" recommendation.

Reviewer #1: All comments have been addressed

Reviewer #2: All comments have been addressed

Reviewer #3: All comments have been addressed

Reviewer #4: All comments have been addressed

2. Is the manuscript technically sound, and do the data support the conclusions?

Reviewer #1: Yes

Reviewer #2: Yes

Reviewer #3: Yes

Reviewer #4: (No Response)

3. Has the statistical analysis been performed appropriately and rigorously? 

Reviewer #1: Yes

Reviewer #2: Yes

Reviewer #3: Yes

Reviewer #4: (No Response)

4. Have the authors made all data underlying the findings in their manuscript fully available?

Reviewer #1: Yes

Reviewer #2: Yes

Reviewer #3: Yes

Reviewer #4: (No Response)

5. Is the manuscript presented in an intelligible fashion and written in standard English?

Reviewer #1: Yes

Reviewer #2: Yes

Reviewer #3: Yes

Reviewer #4: (No Response)

6. Review Comments to the Author

Reviewer #1: All comments have been addressed. The publication criteria have been met. I have no further comment.

Reviewer #2: (No Response)

Reviewer #3: (No Response)

Reviewer #4: (No Response)

7. PLOS authors have the option to publish the peer review history of their article (what does this mean?). If published, this will include your full peer review and any attached files.

Reviewer #1: No

Reviewer #2: **Yes: **Amos Ssematimba

Reviewer #3: No

Reviewer #4: No

---

## [Editor Report · Acceptance letter]

PONE-D-25-03162R1

PLOS ONE

Dear Dr. Jung,

I'm pleased to inform you that your manuscript has been deemed suitable for publication in PLOS ONE. Congratulations! Your manuscript is now being handed over to our production team.

Kind regards,

on behalf of

Dr. Eunha Shim

Academic Editor

PLOS ONE